# Nanodrug Delivery Systems for Myasthenia Gravis: Advances and Perspectives

**DOI:** 10.3390/pharmaceutics16050651

**Published:** 2024-05-11

**Authors:** Jiayan Huang, Zhao Yan, Yafang Song, Tongkai Chen

**Affiliations:** Science and Technology Innovation Center, Guangzhou University of Chinese Medicine, Guangzhou 510405, China; 20211110940@stu.gzucm.edu.cn (J.H.); 20221111006@stu.gzucm.edu.cn (Z.Y.)

**Keywords:** myasthenia gravis, nanomedicine, ferroptosis, immunomodulation, mitochondrial dysfunction

## Abstract

Myasthenia gravis (MG) is a rare chronic autoimmune disease caused by the production of autoantibodies against the postsynaptic membrane receptors present at the neuromuscular junction. This condition is characterized by fatigue and muscle weakness, including diplopia, ptosis, and systemic impairment. Emerging evidence suggests that in addition to immune dysregulation, the pathogenesis of MG may involve mitochondrial damage and ferroptosis. Mitochondria are the primary site of energy production, and the reactive oxygen species (ROS) generated due to mitochondrial dysfunction can induce ferroptosis. Nanomedicines have been extensively employed to treat various disorders due to their modifiability and good biocompatibility, but their application in MG management has been rather limited. Nevertheless, nanodrug delivery systems that carry immunomodulatory agents, anti-oxidants, or ferroptosis inhibitors could be effective for the treatment of MG. Therefore, this review focuses on various nanoplatforms aimed at attenuating immune dysregulation, restoring mitochondrial function, and inhibiting ferroptosis that could potentially serve as promising agents for targeted MG therapy.

## 1. Introduction

Myasthenia gravis (MG) is a rare autoimmune disease that is driven by B cells, dependent on T cells, and mediated by complement activation and antibodies. It is caused by the production of autoantibodies against acetylcholine receptor (AChR), muscle-specific kinase (Musk), and low-density lipoprotein receptor-related protein 4 (LRP4) expressed on the postsynaptic neuromuscular junctions (NMJs) of skeletal muscle [1]. The binding of these antibodies to postsynaptic receptors causes the inactivation of motor signals. Subsequently, complement activation and the formation of the membrane attack complex decrease the density of AChR expression. Together, these changes impede normal neuromuscular signal transmission, causing the development of MG [2,3]. The global prevalence of MG is 150–250 cases per million population, and this condition has an annual incidence of 8–10 cases per million population years [4]. In China, the incidence of MG is approximately 0.68 per 100,000 people, and the annual hospital admission mortality is 14.69‰ [5]. Notably, the increased prevalence of MG has bolstered further research in this field [6].

Studies have shown that more than 80% of patients with MG are positive for anti-AChR antibodies, and only a minority of MG patients test negative for these antibodies [7,8]. The clinical manifestations of MG include muscle weakness and recurrent fatigue. Moreover, its symptoms often become aggravated after intense activity and can be alleviated by rest [9]. Interestingly, some patients with MG also experience thymus dysfunction [10]. The initial symptoms of MG typically affect the eye, causing diplopia and ptosis [11]. However, the disorder gradually progresses, affecting muscles throughout the body, and it can even become life-threatening in some cases [12]. Overall, the pathogenesis of MG is complicated, and the mechanisms are not clearly understood at present. Thus, most treatment modalities for MG target its symptoms.

Mitochondria, called the powerhouse of eukaryotic cells, are double-membrane-bound cell organelles that play an essential role in cellular metabolism, proliferation, and survival. Mitochondrial morphology and distribution vary across different tissues and organs due to differences in the demand for Adenosine 5′-triphosphate (ATP), which tends to be higher in the brain, heart, and muscles. Moreover, mitochondria are multifunctional organelles involved in the production of bioenergy, biosynthesis, and signal transduction, and they maintain cellular homeostasis by regulating cellular metabolism, stress responses, and adaptive nuclear gene expression during both biological and environmental changes [13,14]. Mitochondrial signal transduction ensures organismal adaptation, underscoring the importance of mitochondrial biology in human health [15]. In addition, mitochondria are highly dynamic and play a vital role in maintaining cellular energy metabolism by producing energy in the form of ATP via oxidative phosphorylation (OXPHOS) through the electron transport chain (ETC) [16]. Furthermore, through fusion and fission, mitochondria form dynamic tubular networks to respond to external stress by eliminating dysfunctional mitochondria while producing healthy ones so as to maintain regular function and prevent disease [17].

Ferroptosis, a unique non-apoptotic form of regulated cell death, was first discovered a decade ago by Dixon et al. [18]. Since then, ferroptosis has been found to possess unique morphological, biochemical, and genetic features that are distinct from those of apoptosis, autophagy, necroptosis, and pyroptosis [19]. The overload of iron and lipid peroxidation are the two primary biochemical processes regulating ferroptosis [20]. In addition, oxidative stress caused by excess ROS can also cause lipid peroxidation, which directly or indirectly induces ferroptosis [21,22]. The typical ultrastructural features of ferroptosis include reduced or disappearing mitochondrial cristae, a ruptured outer membrane, and an increase in membrane density [23]. Ferroptosis involves a complex regulatory network and contributes to the development of various disorders, including neurodegenerative diseases [24], cancer [25], cardiovascular disease [26], and some autoimmune diseases [27]. Ferroptosis has emerged as a focus and hotspot of clinical research as it can be targeted with antioxidants and deferramine [28,29,30]. Thus, ferroptosis is considered a promising therapeutic target for various diseases.

Nanotechnology, a rapidly emerging field of application, provides a mass of new opportunities for medical science and disease management [31]. It presents several advantages with respect to administration and can produce excellent therapeutic effects, attracting the widespread attention of researchers. Nano-sized drug delivery systems consisting of nanoparticles sized < 0.1 µm show enhanced bioavailability and bioactivity. Moreover, they reduce the required drug dosage and can minimize drug-induced adverse effects [32]. Several nano-delivery systems, such as liposomes, dendrimers, lipid nanoparticles [33], nanogels [34], polymeric nanoparticles [35], nanoemulsions, and microemulsions [36], have been developed for health management. Nanotechnology has been employed to treat a variety of diseases, including cancer [37], cardiovascular disease [38], neurodegenerative disorders [39], and immune dysfunction [40], and it has demonstrated exciting prospects for disease diagnosis and treatment. Moreover, nanoparticles can be modified using functional molecules to achieve precisely targeted drug release and improve drug availability [41,42]. In addition, nanoparticles show adequate biocompatibility and degradability and can serve as excellent carriers for several treatment agents [43], including RNA [44] and drugs with poor water solubility and bioavailability [45]. Therefore, nanotechnology-based drug delivery systems appear to be promising for the management of various diseases.

The application of nanotechnology in MG management needs to be expanded. Thus, the present review summarizes the known pathogenic mechanisms of MG, including immune disorder, mitochondrial dysfunction, and ferroptosis. In addition, it focuses on the application of nano-sized drug delivery systems for MG treatment, thereby emphasizing the intrinsic immunomodulatory potential of nanomedicines. Furthermore, this review explores the future prospects of ferroptosis- and mitochondrial-targeted nanotherapeutic agents for MG. Overall, the review highlights the current achievements in this area and proposes novel strategies for improving therapeutic efficiency and promoting the clinical transformation of nanomedicine-based MG treatments.

## 2. Pathogenesis of MG

### 2.1. Immunoregulatory Defects in MG

The mechanism of autoimmune dysfunction in MG is yet to be elucidated. However, defects in immune regulation play a vital role in MG pathogenesis in patients with anti-AChR antibodies. Many studies have implicated inactivated T cells, B cells, and plasma cells in the initiation and sustained production of anti-AChR autoantibodies, which target the NMJ and cause muscle weakness [46]. A normal thymus is crucial for T-cell differentiation and the establishment of central tolerance. It enables the elimination of autoreactive T cells and allows self-tolerant T cells to continue differentiating [47,48]. Under normal physiological conditions, the quantity of B cells is extremely small. However, in most patients with MG, the thymus shows structural and functional alterations, forming germinal centers (GCs) and exhibiting hyperplasia with a large number of B cells [49,50]. The existing B cells, which produce autoantibodies, rely on their interaction with CD4^+^ T cells (including T helper (Th) lymphocytes) to secrete inflammatory cytokines and cause autoimmune responses against self-antigens. This results in the development of MG [51,52]. Several studies have demonstrated that the increase in Th17 cells and dysregulated T regulatory cells (Tregs) is involved in the pathogenesis of MG [53]. Th17 and Treg cells regulate opposite directions of the immune response, and their imbalance causes autoantibody generation and promotes pathogenic autoimmunity [54]. Recent studies have shown that Th17 cells play an important role in mediating chronic inflammation and blocking immune tolerance, while Tregs promote immune tolerance by inhibiting the activation and multiplication of CD4^+^ T cells [55,56]. Th17 cells can release interleukin-17 (IL-17), which indirectly stimulates B cells to produce antibodies or affects the Th1- and Th2-related cytokine balance during neuromuscular transmission in MG patients [57,58,59]. IL-10 and transforming growth factor (TGF)-β are two important suppressor cytokines produced by Tregs, and their downregulation can decrease immune tolerance [60]. Notably, the loss of forkhead box protein P3 (FOXP3) expression in Tregs is also a key contributor to MG, and its downregulation is correlated with the severity of MG [61]. In fact, most studies report a decrease in Tregs or the absence of immunosuppression in MG patients [62]. Although one study found no significant difference in peripheral Treg populations between MG patients and healthy controls, it demonstrated that Tregs isolated from healthy controls can improve the suppression of AChR-activated Treg cells [63]. Some studies have also shown that regulating the Th17/Treg immune imbalance can effectively control MG pathogenesis [64,65]. Therefore, strategies to increase the number of Tregs and downregulate Th17 inflammatory cytokines could be promising for MG treatment. Figure 1 illustrates the immune-mediated pathogenesis of MG.

### 2.2. Pathological Role of Mitochondria in MG

Although MG is currently considered an autoimmune disease involving postsynaptic AChR at the NMJ, mitochondrial damage also plays an important role in the pathogenesis of this disease [66,67]. Indeed, mitochondria have emerged as crucial players in the regulation of cellular homeostatic responses involving energy production, ROS generation, and calcium signaling [68]. Skeletal muscle, the main tissue involved in MG, requires abundant energy to maintain muscle contractions [69]. Mitochondrial function and signal transduction play a key role in energy metabolism during excitation-contraction coupling in mammalian skeletal muscles [70]. Energy metabolism is fundamental to mitochondrial biogenesis and also acts as an important regulatory mechanism in MG [71]. Mitochondrial biogenesis and dynamics enable the formation of an intricate regulatory network to maintain mitochondrial homeostasis, and these are essential as response stressors in the skeletal muscle [72]. Several studies have established the close link between mitochondrial dysfunction and low skeletal muscle mass [73,74]. Mitochondrial dysfunction can reduce skeletal muscle mass and function in Drosophila, increasing the risk of multisystemic failure [75]. Catalase expression in muscle mitochondria can efficiently prevent mitochondrial dysfunction and NMJ disruption and modulate skeletal muscle atrophy and weakness [76]. Our previous studies confirmed that experimental autoimmune myasthenia gravis (EAMG) rats exhibit ultrastructural damage to muscle mitochondria, including abnormal mitochondrial biogenesis, fusion, and fission [77,78]. In addition, we found that the mitochondrial dynamics- and biogenesis-associated factors in peripheral blood mononuclear cells (PBMCs) may act as potential biomarkers for the diagnosis of MG [79]. Furthermore, the important role of mitochondrial autophagy in myocytes, mediated by the PINK1/Parkin pathway, has also been confirmed [80].

The accumulation of mitochondria is the most obvious in oxidative muscle fibers. Hence, muscle mitochondrial biogenesis plays a compensatory role under energetically compromised conditions [81]. The alterations in mitochondrial biogenesis and oxidative activity induced by vitamin D in skeletal muscle cells indicate that the ability of mitochondria to produce ATP is closely related to muscle mass and function [82]. The disrupted balance between mitochondrial fusion and fission in MG results in reduced mitophagy and the accumulation of dysfunctional organelles [83]. Mitochondrial regulation also affects muscle mass and contributes to metabolic diseases [84]. Moreover, the overexpression of peroxisome proliferator-activated receptor-gamma coactivator-1 alpha (PGC-1α), a regulator of mitochondrial biogenesis, can correct neuromuscular dysfunction in cachectic skeletal muscle cells [85]. Furthermore, sera from MG patients can alter the bioenergetic metabolism of mitochondria, enhancing oxidative metabolism and glycolysis [86].

The process of mitophagy can improve skeletal muscle function by increasing mitochondrial density and improving autophagic recycling [87]. Likewise, enhanced tricarboxylic acid cycle activity and mitochondrial fission or fusion also enable the adaptation of energy metabolism and ultimately decrease oxidative stress [88]. Mitochondria are highly plastic and possess good antioxidant capacity, which can promote the functional recovery of injured skeletal muscle under the modulatory effects of cytokines [89]. The antioxidant selenium can help in maintaining the density of mitochondria and limit ROS signals, modulating mitochondrial biology and improving muscular health [90]. Mitochondrial fission increases after muscle injury, leading to the induction of OXPHOS for maintaining normal mitochondrial dynamics and promoting muscle regeneration [91]. In addition, the activation of mitochondrial dynamics by adenosine monophosphate-activated protein kinase (AMPK) can ameliorate mitochondrial dysfunction and facilitate metabolism in skeletal muscles [92]. Interestingly, a previous study suggested that mitochondrial transplantation can promote muscle regeneration and the restoration of muscle function after injury [93]. Together, these findings demonstrate that the complicated regulatory networks that control mitochondrial function present valuable therapeutic opportunities to enhance muscle performance and improve muscle health and well-being. Figure 2 illustrates the mechanisms of mitochondrial damage and ferroptosis in MG.

### 2.3. Potential Pathogenic Effects of Ferroptosis on MG

Iron is an essential trace element with crucial biological functions in the human body. In recent years, ferroptosis has received immense attention in various fields, such as neurodegenerative disease [94], cancer [95], and cardiovascular disease [96]. Under physiological conditions, more than 70% of the body’s iron is used for the synthesis of heme proteins. The rest is mainly found in the liver and skeletal muscle, and other tissues only contain minimal levels of iron [97]. The role of ferroptosis in skeletal muscle disorders has gradually aroused attention, and ferroptosis has been implicated in the progression of sarcopenia [98], rhabdomyolysis (RM) [99], amyotrophic lateral sclerosis (ALS) [100], and Friedreich’s Ataxia (FRDA) [101]. One study showed that serum iron levels are reduced in iron-insufficient MG patients, and these levels are negatively related to the levels of anti-AChR antibodies and IL-6 [102]. Another study demonstrated the negative correlation between serum iron status and muscle mass, providing new insights into the prevention and treatment of muscle loss [103].

Evidence suggests that iron metabolism indicators may be promising for the assessment of disease severity and monitoring of clinical efficacy in patients with MG [104]. In mammalian cells, iron is required for adequate mitochondrial biogenesis and metabolism, and iron deficiency can cause iron homeostasis disorders [105]. Mitochondria are the main sites for iron utilization and accumulation [106], and iron imbalances not only promote ROS production and the consequent oxidative stress but also cause mitochondrial dysfunction and ultimately produce negative effects on health [107,108]. Additionally, excessively high iron levels can also promote ROS production and reduce self-protective autophagy, causing the death of L6 skeletal muscles [109]. However, ferroptosis inhibitors can attenuate mitochondrial lipid peroxidation and morphological alterations in muscle cells [110]. Transferrin receptor protein 1 (Tfr1) deficiency in skeletal muscle also leads to abnormal lipid and iron metabolism, impairing muscle function [111]. Older mice show increased levels of iron in skeletal muscles, which are associated with increased lipid peroxidation. However, oral iron chelators do not redistribute iron, highlighting the complexity and importance of iron homeostasis [112]. Therefore, more efforts and further investigations are warranted to reduce mitochondrial ferroptosis and ultimately promote the body’s health. Figure 3 depicts the distribution of iron in the human body.

## 3. Therapeutic Potential of Nano-Biomedicines in MG

### 3.1. Potential Role of Nano-Biomedicines in Immunomodulation

Currently, there is no specific drug for the treatment of MG. Hence, the treatment strategies are mainly focused on symptom management. Cholinesterase inhibitors, immunosuppressants, and glucocorticoids are commonly used in clinical practice for MG management [113]. Cholinesterase inhibitors can quickly relieve clinical symptoms; however, they have no immunoregulatory effects. Long-term immunosuppressants can also control symptoms, but they cause a high number of toxic adverse effects, make patients prone to drug dependence, and increase the risk of infection and tumor development. Furthermore, some patients even show resistance to conventional immunosuppressants.

Although the extensive application of immunomodulators in clinical practice has improved MG symptoms, it has also resulted in significant adverse effects. Therefore, strategies for reducing drug doses and increasing tissue targeting are warranted. Fortunately, nano-drug delivery systems may serve as desired therapeutic tools for MG [114]. Recently, extracellular vesicles encapsulating with caspase-1 inhibitor were developed for targeted macrophage therapy for EAMG, demonstrating the great potential of nano-biomedicines in MG treatment [115]. The application of nanotechnology has expanded in recent years, especially in the field of medicine (Table 1). For instance, the development of nano-vaccines has not only prolonged antigenic stability and strengthened immunogenicity but also enabled targeted delivery. As a result, nano-vaccines can regulate both humoral and cell-mediated immune responses better than traditional vaccines [116]. Nanotechnology can enhance immunosuppressant activity, reduce their adverse effects and toxicity, and improve their targeted distribution. Nanomaterials showcasing outstanding biocompatibility can act as drug delivery vehicles and have been extensively employed in the field of biomedicine to regulate immunity and decrease the inflammatory response [117].

One study indicated that nanocarriers loaded with immunosuppressants can improve their therapeutic effect and reduce the required drug dose, ultimately achieving drug-specific targeting [118]. Similar findings have also been reported in most other studies. The application of steroidal nano-drugs in Duchenne muscular dystrophy (DMD) can selectively target the dystrophic tissue and significantly reduce adverse effects following long-term treatment [119]. IL-4- and IL-10-conjugated gold nanoparticles can attenuate inflammatory muscle injury through immune regulation, which promotes Treg regeneration and improves muscle function [120]. The immunomodulatory camouflage nano-platform, which has an innovative structure, can effectively evade immune clearance and reduce autoimmune reactions during cancer treatment [121]. In addition, nucleic acid-lipid nano-conjugations that can target immune cells and muscle cells, thereby encouraging muscle growth to achieve sustained therapeutic effects, have also been developed [122]. All these studies demonstrate the potential of nano-delivery systems for the regulation of immune responses in muscle disorders. Immunosuppressants are sometimes unavoidable during MG treatment. The application of nano-drug delivery systems may decrease the frequency and dose of drug treatment, reducing adverse effects such as nausea, vomiting, and diarrhea so as to improve patient prognosis. These nano-immunomodulatory platforms may serve as potential therapeutic tools for MG.

### 3.2. Potential Role of Nano-Biomedicines in Targeted Mitochondrial Therapy

Several studies have shown that boosting muscular mitochondrial function can help in the management of muscle-related diseases. Mitoquinone (MitoQ), a widely-used mitochondria-targeted antioxidant, boosts the use of accumulated muscles to precipitate the balance of energy metabolism in skeletal muscles [123,124]. Moreover, another mitochondrial superoxide scavenger, Mito-TEMPO (MT), also markedly inhibits muscle atrophy by attenuating mitochondrial dysfunction and suppressing inflammatory and oxidative stress factors [125]. Increasing the levels of NAD^+^ precursors can also ameliorate the symptoms of acquired muscle dysfunction by activating mitochondrial metabolism [126,127]. Mitochondria-targeted drugs have been proven highly effective in the management of skeletal muscle-related diseases, but traditional dosage forms exhibit low bioavailability *in vivo*. This problem can be addressed by using nanocarriers. Moreover, nanodrug systems can be easily modified to target specific tissues, which provides a significant advantage in the treatment of muscle disorders. For example, Coenzyme Q10 (CoQ10)-loaded poly (lactic-co-glycolic acid)-poly(ethylene glycol)-triphenylphosphonium bromide nanoparticles (CoQ10-TPP-NPs) were found to obviously increase the activity of the tricarboxylic acid cycle when compared to free-CoQ10 in an in vitro disease model [128]. Further, a nano-system that carried therapeutic genes was modified using a mitochondrial-targeting sequence (MTS) peptide and also showed improved cellular transfection and uptake for the treatment of mitochondrial diseases [129]. In addition, a ROS-responsive mitochondria-targeted liposomal system was exploited to release quercetin (Que), targeting FOXO3A to inhibit oxidative stress-induced inflammation [130]. Triphenylphosphonium (TPP^+^)-hydroxytyrosol (HT), in which TPP^+^ was covalently linked to HT, improved HT accumulation in mitochondria, enhancing mitochondrial function and redox balance [131]. Nanofiber-bundle scaffold-containing nanoparticle-ceria nanozyme (NBS@CeO), an energy-supporting enzyme-mimic nano-scaffold, improved mitochondrial function and regulated the immune microenvironment by eliminating excessive ROS, increasing ATP synthesis, and inducing alternatively activated macrophages (M2) polarization to promote motor function recovery [132]. Therefore, the application of nano-drug delivery systems to improve mitochondrial function may be a feasible strategy for alleviating the symptoms of MG. Interestingly, several nanocarriers have shown excellent biocompatibility in myoblasts/myotubes, with poly(lactic-co-glycolic acid) (PLGA)nanoparticles exhibiting a better interaction effect than mesoporous silica nanoparticles or liposomes [133]. A recent study revealed that nano-positioning and microtubule conformation could correct the spatiotemporal distribution of mitochondria, thereby regulating mitochondrial motility and dynamics [134]. Inevitably, mitochondria-targeting nanomaterials are widely employed in the treatment of cancer. The modification of nanocarriers with mitochondria-targeting moieties, such as novel TPP-based compounds [135] or ROS-responsive systems [136] can only inhibit drug resistance [137] but also induce a synergistic therapeutic effect [138]. These mitochondria-targeting nanomedicines shed light on potential drug design strategies for MG, which warrant deeper exploration. However, it is essential to evaluate the mitochondrial targeting efficiency of nano-drug delivery systems, which may be achieved via cascade targeting through multifunctional modifications.

### 3.3. Potential Effects of Nano-Biomedicines on Ferroptosis

Given that the excess ROS produced by lipid peroxidation is a key contributor to ferroptosis, the synergistic reduction of lipid peroxidation and enhancement of antioxidant defense may be an effective strategy for inhibiting ferroptosis in skeletal muscle cells. Nanotechnology tools that leverage or target ferroptosis have been extensively applied in biomedicine. For example, a nano-system that induces ferroptosis by targeting lipid peroxidation has been leveraged to destroy cancer cells and overcome therapeutic drug resistance due to its natural anti-cancer properties [139,140]. Moreover, recent studies have also indicated that inhibiting mitochondrial ferroptosis and subsequently reducing oxidative stress damage can prevent disease development. Curcumin nanoparticles can effectively cross cellular membranes and show mitochondrial accumulation to effectively suppress erastin-induced ferroptosis in intracerebral hemorrhage [141]. Melanin-like materials combined with antioxidant nano-agents can regulate the production and degradation of radicals to achieve a redox balance, thus preventing ferroptosis-related disorders. These materials also show multifunctional properties and better pharmacokinetics and are thus superior to conventional molecular agents [142]. Ceria-based nanoparticles can prevent mitochondria-dependent ferroptosis by relieving oxidative stress, mitochondrial lipid peroxidation, and membrane potential disruption, thereby reversing myocardial damage and reducing myocardial necrosis [143]. In addition, polydopamine nanoparticles (PDA NPs) can protect against ferroptosis by inhibiting GPX4 ubiquitination and lipid peroxidation and also scavenging mitochondrial ROS to mitigate intervertebral disc degeneration [144]. Similarly, K46Q and carbon quantum dots (QDs)-based mitochondria-targeting nano-composites (TPP-AAV) can inhibit the malonylation of voltage-dependent anion channel 2 (VDAC2) and alleviate the ferroptosis and myocardial dysfunction caused by increased mitochondrial ROS [145]. Thus, nanodrug systems that target mitochondrial ferroptosis by carrying antioxidants or iron-chelating agents could serve as promising therapeutic agents for the treatment of MG. However, the mechanistic contribution of ferroptosis in MG remains unclear. Thus, additional research is required to provide a theoretical foundation for the clinical application of these agents.
pharmaceutics-16-00651-t001_Table 1Table 1Summary of the nanodrug delivery systems used for targeting the immune system, mitochondria, and ferroptosis.TargetDelivery SystemActive Drug/AgentTreatment OutcomesReferencesImmune systemExtracellular vesiclesCaspase-1 inhibitorTargeted macrophages to inhibit the Th17 response and GC response and thereby improve EAMG[115]AuNPsIL-4 or IL-10Shifted the immune response in chronically inflamed dystrophic muscle[146]PLA and nano-HAPDoxycyclineDecreased salivary MMP-8 and plasma IL-1 and TNF-α concentrations[147]Nano-liposomesMPSDecreased serum TGF-β levels and reduced macrophage infiltration in the diaphragm[119]PLGA compositesPolydeoxyribonucleotideRegulated the M1-to-M2 polarization of macrophages and caused immune modulation[148]LNPs and polyplex nanomicellesmRNASupported rapid mRNA expression and a potent immune response[149]LiposomesAlendronateRegulated the M1-to-M2 polarization of macrophages and T-cell functionality[150]Flexible liposome hydrogelDEXReduced joint swelling by increasing macrophage uptake[151]GO nanosheetsGOReversed the dynamic changes to CKs and reduced the activity of Ca^2+^[152]Erythrocyte membrane-camouflaged NPsCD22-shRNA, Aβ aptamersAmeliorated a pro-inflammatory immune environment and could be used to visualize Aβ plaques[153]AuNPsIL-4Directed M2 macrophage polarization and promoted regeneration[154]MitochondriaPLGA NPsSonosensitizer IR780 and ferroptosis activator RSL-3Inhibited the activity of GPX4 and induced ROS generation[155]Lipid-polymer hybrid nano-systemCalycosin and tanshinoneIncreased drug accumulation in cardiac tissue and enabled better infarct size reduction[156]Lipid nanocarrierssiRNA-loaded magnesium phosphate coreReversed mitochondrial dysfunction and alleviated AD neuropathology[157]Ceria NPsAtorvastatinEliminated excessive ROS and protected mitochondrial structure[158]Polydopamine-coated NPsPDA and α-TOSEnabled nanomedicine accumulation in mitochondria to destroy tumor cells[159]Molecularly imprinted polymer NPsMolecularly imprinted polymerBlocked the catalytic activity of DHFR to inhibit DNA synthesis[160]Porous silicon NPsBovine serum albuminDisrupted the mitochondrial respiratory chain[161]PLGA-b-PEG NPsCoQ10Effectively increased the tricarboxylic acid cycle rate[128]LipidosomesQuercetinDecreased ROS generation, increased ATP levels, and enhanced lactate dehydrogenase activity[130]Biomimetic nanocrystalsCurcuminReversed mitochondrial dysfunction, TH^+^ neuron injury, and abnormal α-syn aggregation[162]ZIF-8-coated Prussian blue nanocompositeQuercetinRestored mitochondrial function, restored energy metabolism, and reduced ROS[163]BPNSsMatrineImproved neurotransmitter delivery, removed excess ROS, and decreased neuroinflammation[164]BPNSs-based hydrogelMethylene blueImproved mitochondrial function, and suppressed tau neuropathology[165]Platelet membranes-ICG-SS31-PLGAIndocyanine green and elamipretideReduced mitochondrial oxidative stress, inflammation, and apoptosis[166]FerroptosisPolydopamine NPsPolydopamineDepleted ROS, chelated iron, and inhibited the ubiquitination of GPX4[144]Ceria-based NPsCerium oxideAlleviated oxidative stress and lipid peroxidation and increased GPX4 activity[143]DSPE-PEG 2000 NPsIron oxideRegulated the Beclin1/ATG5-dependent autophagy pathway[167]Melanin NPsMelaninInhibited ROS-related ferroptosis to reduce myocardial injury[168]Poly-PLGA co-polymersAlpha lipoic acidReduced ROS-induced damage and restored heart function.[169]Metal-phenolic nanocomplexesQuercetinAttenuated the free radical burst induced by iron overload and restored iron metabolism homeostasis[170]PDN@AGLAGLDecreased lipid peroxidation, reduced ROS levels, and attenuated ferroptosis[171]MPEG-PTMC NPsCurcuminEnhanced the delivery of Cur to inhibit ferroptosis[141]Polymer NPsResveratrolInhibited ROS generation and excessive accumulation to attenuate ferroptosis[172]PAA@Mn_3_O_4_ NPsMn_3_O_4_Resisted lipid peroxidation and detoxified ROS to suppress ferroptosis[173]NPs: Nanoparticles; Caspase-1: Cysteinyl aspartate-specific proteinase-1; IL: Interleukin; PLA: Polylactic acid; HAP: Hydroxyapatite; AuNPs: Gold nanoparticles; MPS: Methylprednisolone hemisuccinate; TNF-α: Tumor necrosis factor-α; LNPs: Lipid nanoparticles; DEX: Dexamethasone; GO: Graphene oxide; BPNSs: Black phosphorus nanosheets; α-TOS: Alpha-tocopherol succinate; AGL: Apigenin-7-*O*-glucoside; PDN: Nanoparticles; PAA: poly(acrylic) acid.


## 4. Conclusions and Perspectives

This review summarizes the nanodrug delivery systems showing therapeutic potential for the immune dysfunction, skeletal muscle mitochondrial impairment, and ferroptosis seen in MG. The aim of this review was to provide valuable insights into the field of MG drug development and encourage interdisciplinary collaboration to explore nano-sized drug delivery systems. This review highlights how nanodrug delivery systems hold immense potential for the diagnosis and treatment of MG given the accumulating research in this field.

The primary consideration in nanocarrier development is biocompatibility. Some drugs cannot be translated clinically due to the hemolytic reactions and immunogenicity induced by their intravenous injection. The preparation of certain nanocarriers involves toxic chemical agents, which impacts their particle size and potential and increases cytotoxicity. However, the toxicity and effects of nanocarriers in vivo and in vitro still require further investigation. The transition from the preclinical to clinical stage remains a significant challenge due to our limited understanding of how nanomaterials interact with various tissues and organs in the human body. Therefore, modifications to the material and the types of carriers should align with biological requirements to minimize adverse effects.

Targeted surface-modified nanocarriers are crucial for MG treatment. Unmodified nano-drugs can be metabolized by the body. As a result, a limited amount of the active drug reaches the lesion site, significantly reducing drug efficacy. For immunological dysfunction in MG, the homologous targeting of biomimetic nanocarriers can specifically target immune organs or cells, reducing autoantibody production. Recently, there has been a surge in the development of mitochondria-targeted nanomaterials, and considerable therapeutic effects have been achieved. Thus, surface-modified compounds with multiple target sites, including skeletal muscle and mitochondria, can be developed and controlled and drug release can be achieved based on changes in the body’s microenvironment. Thus, nanodrug delivery systems offer unique capabilities, increasing drug efficiency while reducing off-target actions.

Long-term biosafety is also a vital evaluation criterion for nanomedicines. MG is a chronic autoimmune disease, and extended medication regimens are often required for its control. While immunosuppressors have been extensively used in clinical MG treatment, short-term symptom improvement can often be followed by disease exacerbation and increased susceptibility to infections or cancer. Although nanodrug delivery systems have shown good biosafety in skeletal muscle diseases, metabolic pathways in vivo or in vitro remain limited. The continuous monitoring of serum inflammation indicators and organ function is necessary after nanodrug administration. Thus, additional research is needed to explore the metabolic processes and mechanisms through which nanocarriers improve the health of skeletal muscle.

Recently, several drugs with different mechanisms of action have been approved by the FDA. These drugs are gradually entering MG clinical trials that assess their tolerability, safety, immunogenicity, pharmacokinetics, and pharmacodynamics. These novel targeted biologics will undoubtedly increase the treatment opportunities for MG patients. Emerging treatment agents and strategies, such as neonatal Fc receptor inhibitors, complement inhibitors, and regenerative therapy, offer feasible approaches for MG treatment [174,175]. With more in-depth research into MG, novel targeted biologics are gradually progressing towards clinical translation (Table 2). Current studies have indicated that nanodrug delivery systems can improve drug pharmacokinetics and enable targeted therapy, potentially improving the bioavailability of targeted biologics and allowing accurate drug release to enhance treatment efficacy and limit adverse effects. Therefore, the application of nanotechnology in the field of biomedicine could pave the way for the drug development to treat MG.

In conclusion, nanomedicines exhibit potential advantages in immune regulation and can preserve mitochondrial function. However, the challenge of clinical translation must be addressed. While nanotherapies for MG are in their early stages, prioritizing the design of nanodrug delivery systems based on the pathogenesis of MG could enable the development of personalized and optimized clinical treatment strategies. Undoubtedly, nanomaterial-based drug delivery may offer more accurate and effective clinical outcomes for MG in the near future.

## Figures and Tables

**Figure 1 pharmaceutics-16-00651-f001:**
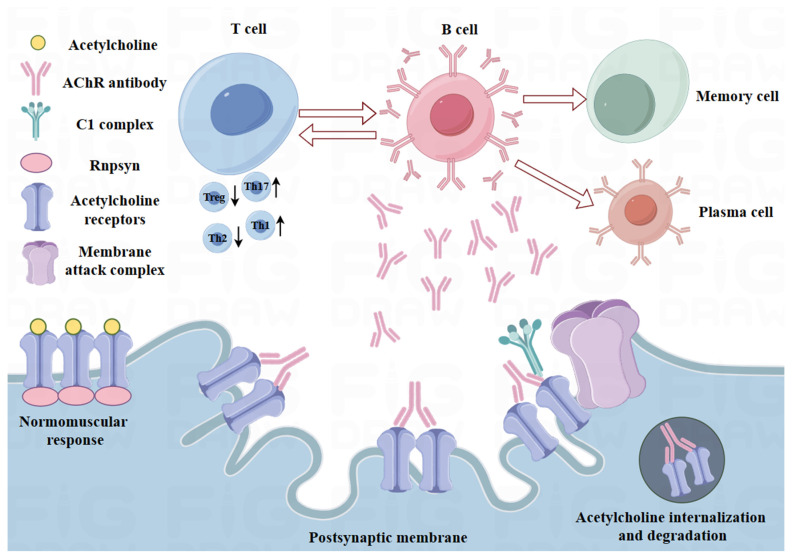
Schematic showing the pathogenesis of acetylcholine receptor-related myasthenia gravis (AChR-MG). T cells interact with B cells, which further differentiate into plasma and memory B cells. The antibodies produced by plasma cells bind to the AChR in the postsynaptic membrane forming the membrane attack complex after binding to the complement, which results in the internalization and degradation of AChR and prevents normal neuromuscular transmission. Created using Figdraw2.0 (www.figdraw.com, accessed on 30 April 2024).

**Figure 2 pharmaceutics-16-00651-f002:**
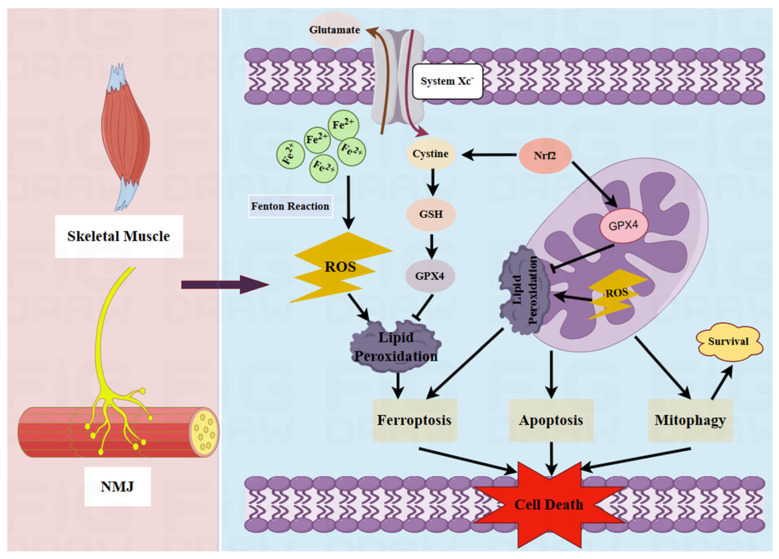
Mechanisms of mitochondrial damage and ferroptosis in MG. Oxidative stress in mitochondria leads to the production of ROS. This can increase membrane potential, mitochondrial swelling and lipid peroxidation, the dysregulation of mitochondrial dynamics, cytochrome C activity, and autophagy, resulting in apoptosis. Moreover, the ROS generated by excessive Fe^2+^ and the Fenton reaction also induce lipid peroxidation, promoting ferroptosis. Nuclear factor erythroid 2-related factor 2 (Nrf2) can upregulate glutathione peroxidase 4 (GPX4) to inhibit the production of ROS, thereby preventing lipid peroxidation and inhibiting cell death. Created using Figdraw 2.0 (www.figdraw.com, accessed on 30 April 2024).

**Figure 3 pharmaceutics-16-00651-f003:**
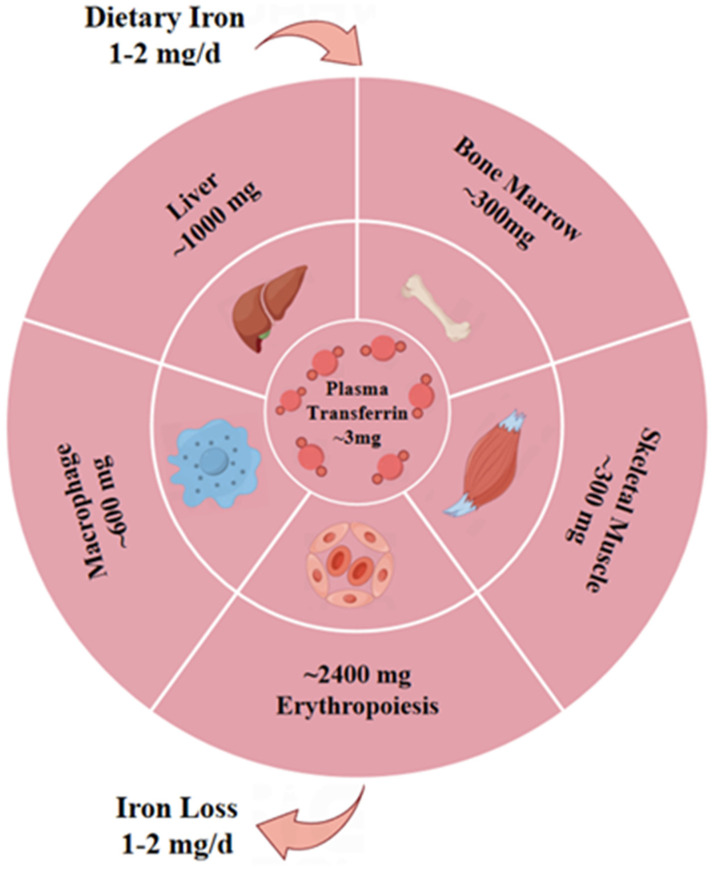
Schematic diagram showing the distribution of iron in the human body. Around 60–75% of the body’s iron is present as part of hemoglobin and in the macrophages of the reticuloendothelial system. Skeletal muscle accounts for a significant 7–8% of the body’s iron, and 20–30% of the iron is stored in the liver. The daily absorption and loss of iron is about 1–2 mg. Created using Figdraw 2.0 (www.figdraw.com, accessed on 30 April 2024).

**Table 2 pharmaceutics-16-00651-t002:** Clinical trials focused on neonatal Fc receptor inhibitors, complement inhibitors, IL-6R-targeting agents, CAR-T therapy, and B cell-targeting agents registered on clinicaltrials.gov in 2024.

Treatment Strategy	NCT Number	Drug	Actual Enrollment	Age	Phase	Status
Antagonize neonatal Fc receptor	NCT05681715	Rozanolixizumab	62	≥18	Phase 3	On going
	NCT04951622	Nipocalimab	198	≥18	Phase 3	Recruiting
	NCT05265273	Nipocalimab	12	2~17	Phase 2 Phase 3	Recruiting
	NCT05403541	Batoclimab	240	≥18	Phase 3	Recruiting
	NCT04980495	Efgartigimod	69	≥18	Phase 3	On going
	NCT05374590	Efgartigimod	12	2~18	Phase 2 Phase 3	Recruiting
	NCT04833894	Efgartigimod	12	2~18	Phase 2 Phase 3	Recruiting
	NCT04818671	Efgartigimod	183	≥18	Phase 3	On going
Inhibit complement	NCT06055959	Zilucoplan	8	12~17	Phase 2 Phase 3	Recruiting
	NCT04225871	Zilucoplan	200	≥18	Phase 3	On going
	NCT05514873	Zilucoplan	26	18~85	Phase 3	On going
	NCT05644561	Ravulizumab	12	Not limited	Phase 3	Recruiting
	NCT05070858	Pozelimab and Cemdisiran	235	≥18	Phase 3	Recruiting
	NCT06282159	DNTH103	60	18~75	Phase 2	Recruiting
Target IL-6R	NCT05067348	Tocilizumab	64	18~80	Phase 2	Recruiting
	NCT05716035	Tocilizumab	64	18~80	Phase 2 Phase 3	Recruiting
	NCT04963270	Satralizumab	185	≥12	Phase 3	Recruiting
CAR-T cells	NCT05828225	CD19 CAR-T cells	9	≥18	Phase 1	Recruiting
	NCT04146051	Descartes-08	30	≥18	Phase 2	Recruiting
Target B cells	NCT04524273	Inebilizumab	238	≥18	Phase 3	On going
	NCT05737160	Telitacicept	100	18~80	Phase 3	Recruiting

## Data Availability

No new data were created or analyzed in this study. Data sharing is not applicable to this article.

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
