# Peer review of "Nanodrug Delivery Systems for Myasthenia Gravis: Advances and Perspectives"

_pharmaceutics, 2024, doi:10.3390/pharmaceutics16050651_

Round 1
Reviewer 1 Report
Comments and Suggestions for Authors
Considering the severity of Myasthenia Gravis (MG) disease pathology and the limited therapeutic options available to the affected patients, searching new therapeutic agents and the delivery systems appear to be a timely approach. Nanotechnology, nanoparticles and drug delivery platforms using nanotechnology-based approach are evolving in a steady rate to tackle many life-threatening diseases including managing neurological complications.
In the present review article, the authors made commendable attempt in summarizing the pathogenesis of MG, molecular basis of MG development and immunobilogical role in disease prohression, mitochondrial contribution, ROS-mediated signaling events and associated ferroptossis linked to MG, therapeutic potential of nanobiomedicines to treat MG with a focus of Mitochondria-specific targeted therapy that interferes with ferrroptosis. They have provided a comprehensive table 9Table 1) summarizing delivery system, pharmacological intervention (active drug/agent) and treatment modalities pulling from an exhaustive literature search which definitely will be helpful to the MG researchers and the clinical providers to understand and manage the disease.
1. I would recommend adding a dedicated table and discussion section listing and describing past, ongoing and futre clinical trials (with the CT registration number) and the goal and finding of the trials using nanomedicines to treat MG patients. Recently investigational new drug CNP-10 received 6FDA ND clearance for Cour’s myasthenia gravis therapy trial. The trial will assess the tolerability, safety, immunogenicity and initial efficacy of the drug CNP-10. Alo include the recent phase 2 clinical trial results of Nipocalimab (https://classic.clinicaltrials.gov/ct2/show/NCT04951622) and the planned Pahes 3 clinical trial activity with this immunotherapeutic drug. Also include discussion on Phase 3 clinical trial of Efficacy and Safety Study of ARGX-113 in Patients with MG Who Have Generalized Muscle Weakness (ADAPT).
2. I also suggest listing and explaining all the abbreviations and acronyms use in the current manuscript I a table at the beginning of the text to orient the readers with the terminologies to ease the effort of reading.
Reviewer 2 Report
Comments and Suggestions for Authors
A neuromuscular junction disorder Myasthenia Gravis (MG) is not common desease, but for thousands of patients worldwide the proper and effective treatment can relieve symptoms and improve quality of life. Nanotechnology is a modern therapeutic approach with many prospects and strengths. The presented review of “Nanodrug delivery systems for myasthenia gravis” seems to be very important and timely. The review was completed at a good scientific level and can be published after correcting some comments.
The manuscript is clear and relevant, but there are some structuring issues. For example, the authors combine cholinesterase inhibitors into the section of immunomodulators, which is not entirely correct. Indeed, today the drugs of choice for MG are cholinesterase inhibitors; their description should be placed in a separate subsection. Immunomodulators (suppressants in the case of MG) occupy a large niche of nanomedicines and can be described separately. Attention should be paid to the successes of nanomedicine for prevalent autoimmune disorders (DOI: 10.1016/j.addr.2024.115194). The authors consider nanovaccines as an example of the success of nanoparticular immunomodulators; so, they should evaluate cerium oxide nanoparticles, which can be tuned either for vaccine adjuvanticity or drug delivery, to reduce autoimmune inflammatory processes (including targeted macrophage polarization and mitigation of cytokine storm, see DOI: 10.1016/j.ejmcr.2024.100141).
In the section on the therapy of ferroptosis in MG, it is incorrect to consider inducers of ferroptosis; this applies to the therapy of other diseases (for example, oncological or infectious). If ferroptosis is involved in the progression of MG, then MG therapy requires reducing its effects, and not vice versa. Authors should revise the Table 1, remove irrelevant references and add more relevant ones, too. For example, poly(acrylic) acid coated Mn3O4 nanoparticles inhibit ferritinophagy-mediated iron mobilization and detoxify ROS, which collectively confer the prominent inhibition of ferroptosis (DOI: 10.1038/s41467-023-43308-w).
Taking into account the previous comment about the relevance of ferroptosis cited works, all other references are relevant, recent (almost all are not older than 5 years) and do not include an excessive number of self-citations (all self-citations are justified).
The drawings are appropriate, but Fig. 4 is duplicated as a graphical abstract; it is proposed to be removed from the text body. Authors must provide a link to the graphic software used (as an example, see Fig.2 DOI: 10.1038/s41392-023-01409-4).
The Conclusion needs to be revised to make it more expressive. Since the title of the review contains the inscription “Advances and Perspectives”, Authors should make a broader conclusion about the possible prospects for the development of nanomedicine in the treatment of the disease being reviewed.
Additional minor comments are in the Table.
|
line |
text |
Comment |
|
76 |
Ferroptosis, a novel form of programmed cell death, was first proposed a decade ago by Dixon et al [18]. |
Iron-dependent oxytosis is NOT a new form of programmed cell death, it was only recently discovered and called “Ferroptosis” by Stockwell and Dixon. Sentence needs to be rephrased. |
|
153 |
Fig.1, Picture in a circle and text in the lower right corner. |
The caption is not clear. Does anti-AChR antibody binding to AChR prevent internalization and degradation of acetylcholine? Or does this binding lead to antibodies internalization and degradation? Or internalization and degradation of AChR? Clarify. |
|
158 222 258 |
Created using Figdraw |
It is recommended to provide a link (www.figdraw.com). |
|
317 |
NBS@CeO |
All abbreviations must be expanded upon first mention in the text. Check the entire manuscript carefully. Moreover, the cited article [133] deals with cerium oxide nanoparticles, a very promising compound for the purposes considered by the authors of the review. These nanoparticles require closer attention (see above). |
|
349 352 |
PDA TPP-AAV, VDAC2 |
Abbreviations must be expanded. |
|
364 |
Table Biomimetic nanozymes [144]
|
Biomimetic nanozymes themselves cannot be a delivery system. In this case, the carrier and the active drug are the same compound, namely ceria-based nanoparticles. |
|
364 |
Table [169] |
Fe3+-doped MnO2 nanosheets were designed to induce ferroptosis. They cannot be used for MG therapy; their function is the opposite (the programmed cell death). |
|
364 |
Table [171] |
The same comment. Ferric-loaded lipid nanoparticles induce ferroptosis. |
|
364 |
Table [172] |
The same comment. The cited paper deals with ferroptosis-inducing nanosystem for tumor immunotherapy, it is unlikely to be applicable for the treatment of autoimmune diseases like MG. |
Reviewer 3 Report
Comments and Suggestions for Authors
The article presented by Yan et al. presented a study about Myasthenia gravis (MG), an autoimmune disease, where autoantibodies target neuromuscular junction receptors, leading to fatigue and muscle weakness. Recent research suggests involvement of mitochondrial damage and ferroptosis. While nanomedicines, known for their adaptability and biocompatibility, have seen limited use in MG treatment, nanodrug delivery systems carrying immunomodulators, antioxidants, or ferroptosis inhibitors show promise in managing MG by addressing immune dysregulation, restoring mitochondrial function, and inhibiting ferroptosis.
The review article is well-written with appropriate sections covering an introduction to the disease, its relevance, and general symptomatology. It effectively delves into the pathogenesis, providing detailed explanations of the underlying mechanisms. However, when discussing treatments, the authors could enhance the potential of the article by providing more critical analysis. For instance, while Table 1 offers a concise summary, there's a need to expand on the content within the text, elucidating why these studies are truly relevant. Additionally, there should be a discussion on which treatments are currently utilized, which are not, and a critical assessment of their efficacy in improving patient outcomes.
The conclusions, while comprehensive, could be more direct, aligning closely with the insights presented throughout the article.
The references utilized are appropriate.
Therefore, this reviewer suggests that this paper could be considered for publication in Pharmaceutics after major modifications.
Round 2
Reviewer 3 Report
Comments and Suggestions for Authors
The authors have made the suggested modifications. This reviewer proposes that the article be accepted.